# DCIS and LCIS: Are the Risk Factors for Developing In Situ Breast Cancer Different?

**DOI:** 10.3390/cancers15174397

**Published:** 2023-09-02

**Authors:** Jasmine Timbres, Kelly Kohut, Michele Caneppele, Maria Troy, Marjanka K. Schmidt, Rebecca Roylance, Elinor Sawyer

**Affiliations:** 1Breast Cancer Genetics, King’s College London, London SE1 9RT, UK; 2St George’s University Hospitals NHS Foundation Trust, Blackshaw Rd, London SW17 0QT, UK; 3Excelya, 35 rue de Paris, 92100 Boulogne, France; 4Guy’s and St Thomas’ NHS Foundation Trust, Great Maze Pond, London SE1 9RT, UK; 5Division of Molecular Pathology, Netherlands Cancer Institute, Plesmanlaan 121, 1066 CX Amsterdam, The Netherlands; 6Department of Clinical Genetics, Leiden University Medical Centre, 2333 ZA Leiden, The Netherlands; 7University College London Hospitals NHS Foundation Trust, 235 Euston Rd., London NW1 2BU, UK

**Keywords:** LCIS, DCIS, case control, breast cancer, risk factor, logistic regression, breastfeeding, hormone replacement therapy, HRT

## Abstract

**Simple Summary:**

Ductal carcinoma in situ (DCIS) is recognised as a precursor to invasive breast cancer (IBC), while lobular carcinoma in situ (LCIS) is considered a risk factor for subsequent IBC. To investigate whether the factors that increase the risks of DCIS and LCIS also predispose to IBC, we analysed risk factors for 3075 DCIS cases, 338 LCIS cases, and 1584 controls aged 35–60. Results showed that breastfeeding after childbirth decreased risks of DCIS and LCIS, similar to the association with IBC. Post-menopausal hormone replacement therapy (HRT) increased the risks of DCIS and LCIS, especially in long-term use (10+ years), with a stronger association with LCIS. However, neither parity nor an increasing number of births decreased the risks of DCIS or LCIS, as they do IBC. The study shows both similarities and differences in the risk factors affecting in situ breast cancer and IBC, and results suggest that regular surveillance is warranted in post-menopausal women taking long-term HRT.

**Abstract:**

Ductal carcinoma in situ (DCIS) is widely accepted as a precursor of invasive ductal carcinoma (IDC). Lobular carcinoma in situ (LCIS) is considered a risk factor for invasive lobular carcinoma (ILC), and it is unclear whether LCIS is also a precursor. Therefore, it would be expected that similar risk factors predispose to both DCIS and IDC, but not necessarily LCIS and ILC. This study examined associations with risk factors using data from 3075 DCIS cases, 338 LCIS cases, and 1584 controls aged 35–60, recruited from the UK-based GLACIER and ICICLE case-control studies between 2007 and 2012. Analysis showed that breastfeeding in parous women was protective against DCIS and LCIS, which is consistent with research on invasive breast cancer (IBC). Additionally, long-term use of HRT in post-menopausal women increased the risk of DCIS and LCIS, with a stronger association in LCIS, similar to the association with ILC. Contrary to findings with IBC, parity and the number of births were not protective against DCIS or LCIS, while oral contraceptives showed an unexpected protective effect. These findings suggest both similarities and differences in risk factors for DCIS and LCIS compared to IBC and that there may be justification for increased breast surveillance in post-menopausal women taking long-term HRT.

## 1. Introduction

Breast cancer is the most common cancer in women, accounting for over 2 million new diagnoses globally each year [1]. Although the incidence has increased over time [2], detection and treatment methods have also improved, leading to improved survival in developed countries [3]. However, breast cancer mortality rates remain high in lower-income countries despite their lower incidence [1,3,4]. Many invasive breast cancers (IBC) are associated with pre-invasive or in situ breast cancer [5,6], which consists of two main histopathological entities: ductal carcinoma in situ (DCIS) and lobular carcinoma in situ (LCIS).

DCIS is a non-invasive form of breast cancer which arises in the ducts of the breast and accounts for about 15% of breast cancer diagnoses in the UK, with over 7000 women diagnosed per year [7]. DCIS is often thought of as pre-malignant, or a precursor to invasive breast cancer, specifically invasive ductal carcinoma (IDC) [8]. However, the mechanisms for DCIS becoming invasive are not entirely understood, and IBC does not always have associated DCIS. Furthermore, not all DCIS develops into IBC, and therefore DCIS has been characterised as a non-obligate precursor [9]. Despite this, DCIS is often treated like IBC; with surgery, with or without radiotherapy, and endocrine therapy. Some women equate DCIS to invasive cancer [9,10], which can lead both to emotional distress and more radical treatment than needed. Previous studies have shown increased fears of recurrence [11,12] or death after DCIS [13,14], and similar levels of anxiety and depression in women diagnosed with DCIS when compared to women diagnosed with stage I-II IBC [13,15]. Studies have estimated that 14–53% of women with DCIS will develop IBC dependent on the type of treatment that the primary DCIS received [9,16,17,18,19,20].

In contrast, LCIS is clinically and radiographically occult and generally an incidental finding on biopsy (~3% of needle biopsies) [21] or associated with invasive lobular carcinoma (ILC). LCIS has often been considered more as a risk factor for invasive disease than as a true precursor [22]; this is because only 7–12% of women with pure LCIS develop subsequent invasive disease [23,24], and only ~23–50% of ipsilateral invasive disease is ILC after LCIS, the remainder being IDC [24,25,26]. There is also an increased risk of contralateral invasive disease after LCIS [27]. However, molecular analysis of co-existing LCIS and ILC shows very similar somatic genetics, suggesting that LCIS can be a precursor of ILC [28,29]. As the risk of IBC following LCIS is much lower than the risk after DCIS (~7% [24] compared to ~30% at 10 years, respectively [18]), the treatment is less radical, with most women just being offered increased surveillance rather than surgery or radiotherapy, with the exception of pleomorphic LCIS [30].

After the introduction of the breast screening programme, the number of women diagnosed with in situ disease in the UK has almost tripled since the 1990s [31]. It is likely that many in situ cases in the population remain undiagnosed, with autopsy studies suggesting that 9% of the female population has DCIS [20,32], while the prevalence of undiagnosed LCIS is unknown.

This study aims to summarise the risk factors that have been previously shown to predispose to DCIS and LCIS and to determine whether the risks associated with these risk factors are similar for DCIS and LCIS, and to associations previously found with IBC. This will be explored using a large cohort of DCIS and LCIS cases recruited to the ICICLE (A study to Investigate the genetiCs of In situ Carcinoma of the ductaL subtypE) and GLACIER (A study to investigate the Genetics of LobulAr Carcinoma In situ in EuRope) studies, which collectively recruited 2121 controls and 5900 cases between 2007 and 2012.

## 2. Review of Risk Factors for Developing In Situ Breast Cancer

Although there have been a number of studies looking at risk factors predisposing to DCIS, the studies looking at LCIS are limited and are likely underpowered to detect significant associations due to small study populations.

### 2.1. Genetic/Inherited Risk Factors

#### 2.1.1. Family History

It is well documented that the risk of IBC is much higher in those with a family history of breast cancer than those without, with a previous study of over 100,000 UK women estimating that 15% of women diagnosed with breast cancer have a first-degree family history of breast cancer, corresponding to a ~1.8-fold increase in breast cancer risk (either in situ or invasive) [33].

A similar association is also seen with DCIS [34,35,36,37,38,39,40,41], even when restricted to oestrogen receptor-positive (ER-positive) DCIS only [41] and when stratified by menopausal status or age [37,39]. In studies looking at both DCIS and LCIS, one study found an increased risk of both DCIS and LCIS with a first-degree family history of breast cancer [42], and other studies found an association with DCIS but not LCIS [35,38,43], which could have been due to small numbers. However, one study did find a slightly larger effect estimate in the association between family history and LCIS (OR: 2.95, 95% CI: 1.65–5.27) than DCIS (OR: 2.68, 95% CI: 1.93–3.72) [42], albeit with overlapping confidence intervals.

#### 2.1.2. Genetic Mutations

The increased risk of IBC due to rare inherited genetic mutations in genes such as *BRCA1*, *BRCA2*, *TP53*, and more recently, *PALB2* and *CHEK2*, is well established. It is understood that the presence of genetic mutations is more prevalent in those diagnosed at a younger age, particularly in oestrogen receptor-negative IBC (ER-negative IBC), with all women in the UK being offered genetic testing if diagnosed with ER-negative IBC aged 60 or under [44]. *BRCA1* and *BRCA2* mutations are found in around 10% of IBC diagnoses and confer a 5–10-fold increase in the odds of developing breast cancer [45]. Rarer mutations such as *PALB2* gene mutations are found in <1% of breast cancers [46], and *TP53* mutations in up to 3% of women diagnosed with breast cancer under 30 [47]. Mutations in the *CDH1* gene have been specifically associated with ILC, often with a history of diffuse gastric cancer [48], and ILC is more common in *BRCA2* carriers than *BRCA1* carriers [49]. However, IDC remains the most common type of breast cancer in those with mutations in the *BRCA1* or *BRCA2* genes [49].

In many countries, women with DCIS do not qualify for genetic testing despite studies (including ICICLE [50]) showing a strong association with *BRCA2* and *CHEK2* mutations and a weaker association with *PALB2*, *BRCA1*, and *TP53*, although the frequency of *BRCA* mutations appears to be less frequent in DCIS than invasive disease [51,52]. Approximately 8–9% of DCIS cases occurring under the age of 50 years have a pathogenic germline genetic mutation [53,54], increasing to 29% in women under 40 with ER-positive DCIS and a family history of breast cancer.

Data on pure LCIS is very limited, but previous research involving cases from the GLACIER study has shown an association between pathogenic germline mutations in the *CHEK2* gene and increased risk of LCIS, although this mutation was rare (3% of LCIS cases) [53]. There was no evidence of an association between *CDH1* mutations and unilateral pure LCIS, but germline *CDH1* mutations have been found in women with bilateral LCIS (with/without ILC) [55].

Over the past decade, over 300 low-risk single nucleotide polymorphisms (SNPs) have been shown to predispose to IBC. A total of 313 of these SNPs have been combined to develop a polygenic risk score (PRS), which shows a greater than 30% increased risk of breast cancer for those that fall into the highest 1% of the PRS [56,57,58,59,60]. Many of the SNPs that form the 313-SNP PRS predispose to both DCIS and LCIS [50,61].

### 2.2. Age

IBC incidence rates increase with increasing age [7]. Similarly, DCIS is more commonly diagnosed in women over the age of 50, with only around 5% of DCIS diagnoses occurring in women <40 [62]. However, this statistic is likely to be affected by breast screening being offered to women >50 in the UK. Higher frequency of DCIS at older ages has been seen in various studies [37,38], including when restricted to ER-positive disease in women of African descent [63], post-menopausal women [64], and women aged <50 at diagnosis [37]. Studies comparing DCIS to LCIS showed a higher frequency of DCIS at older ages compared to LCIS [38].

### 2.3. Reproductive Factors

#### 2.3.1. Parity

There has been a lot of research into the relationship between childbirth and the risk of IBC. Studies have shown that parous women are at decreased risk of developing ER-positive breast cancer and that increasing parity is associated with decreased risk of ER-positive breast cancer, but these associations were not found with ER-negative breast cancer [65,66,67,68].

Some studies have found that the decreased risk was only apparent over a certain number of births, such as three or more births [40,63,69], four or more births [36,70], or even five or more births [71], when compared to nulliparous women. Similarly, studies have also found that being nulliparous can increase risks of developing DCIS [35,37] and that increasing the number of births decreases the risk of developing in situ breast cancer when DCIS and LCIS are considered together [40,42,70], or DCIS alone [38,71,72]. There has not been much research into receptor subtypes of DCIS and their relationship with parity, but one study found no association between the risk of ER-positive DCIS and parity [41], while another found reduced odds of ER-positive DCIS with three or more pregnancies [63].

Due to the small numbers, there is little data on the relationship between parity and LCIS alone. Two studies that included both DCIS and LCIS showed no evidence of an association between LCIS and being parous or parity number [35,38] but did show an association with DCIS.

#### 2.3.2. Age at First Birth

Increasing age at first birth has also been shown to increase the risk of IBC [66,67,73]. Some have suggested that age at first birth is responsible for the association between parity and breast cancer risk, as women with a younger age at first birth are more likely to have a higher parity [74]. It has also been shown that there is an increased risk of breast cancer in the first 5 years after birth [75].

The association between older age at first live birth and higher odds of developing breast cancer is also seen with DCIS [38,39,40,63,70]; however, studies have not found this association with LCIS, possibly due to low numbers [42]. Some research has found this association in specific subsets only, for example, in pre-menopausal women >30 at first birth [39] or in women <50 who were 25 or over at first birth [41]. However, one study found that this association between age at first birth and DCIS was no longer found after adjustment by covariates, including parity, age at menopause, age at menarche, and stratification by BMI [71].

#### 2.3.3. Breastfeeding

Breastfeeding has been found to be protective against IBC [65,76,77]. However, multiple studies have shown no evidence of this association with in situ breast cancer [41,42,63,70], but this could be due to small study samples and, therefore, low power [42,63], racial differences [41], or due to selection bias (including specific occupations only) [70]. Additionally, one study in the US found that for parous women who breastfed for at least 2 years of their life, the risk of DCIS increased [36].

### 2.4. Menstrual Factors

Early menarche and late menopause are associated with an increased risk of IBC due to the increased years of exposure to endogenous hormones, and this association is stronger in ILC compared to IDC [78]. However, age at menarche has not been shown to be associated with the risk of in situ breast cancer [37,38,42,43] when included as a categorical variable, but one study found evidence of a significant trend in age at menarche and decreasing risk of developing in situ breast cancer [40]. Another study found that while the increasing age at menarche was significantly associated with IBC and not DCIS, odds ratio (OR) estimates were in the same direction for DCIS [79], suggesting that there may be a weak association.

There has, however, been evidence of an association between age at menopause and risk of developing DCIS. Previous research has found that women aged 55 or over at menopause had a significantly increased risk of developing in situ breast cancer [40,71], with significant evidence of a trend with increasing age [40]. In contrast, one study found an increased risk of DCIS in women aged <45 at menopause, and that being 45–49 at the time of menopause decreased the risks of developing LCIS, compared to women aged 50–54 at menopause [43].

### 2.5. Lifestyle Factors

Lifestyle behaviours such as smoking and alcohol consumption are modifiable risk factors that have been associated with increased IBC risk [80,81,82,83,84,85,86]. However, there has been little evidence of an association between these behaviours and increased risk of in situ breast cancer. One study found an association between high levels of weekly alcohol intake and risk of DCIS, but not with LCIS [42], and this was not confirmed in other studies [38,43,87]. There is no evidence of an association between smoking and developing in situ breast cancer [38,43,87].

Having a higher BMI has been associated with post-menopausal ER-positive IBC [88], and there is some evidence that it has a stronger association with IDC than DCIS [72]. Some studies have shown a decreased risk of DCIS with high BMI in pre-menopausal women [35,36,37], while in post-menopausal women, some studies found no association between BMI and in situ breast cancer [37,39,42,89]. However, in the UK Biobank study, BMI ≥30 was associated with an increased risk of DCIS [87]. There is no clear association between LCIS and BMI [90].

### 2.6. Exogenous Hormones

Increased hormone exposure can also come from the use of exogenous hormones, such as hormone replacement therapies (HRT) or some forms of contraception.

Various studies have, so far, found evidence of a small increased risk of breast cancer with oral contraceptive use [91]. One study found that oral contraceptives were associated with a small increase in risk of DCIS, which was seen mostly in former users or women without a family history, while the age at which contraceptives were started was not associated with the risk of DCIS [40]. This study did not see any associations between LCIS and contraceptive use. Small increases in the risk of ER-positive DCIS after oral contraceptive use within 10 years have been described, while an increased risk of invasive disease is seen in women taking oral contraceptives for 10 or more years, in women <50 [41]. Other studies found no association between in situ breast cancers and oral contraceptives [92], even when the risk of IBC was increased [42]. No significant difference in risk of DCIS was seen in women using hormonal intrauterine contraceptive devices [93].

However, previous studies have shown clear evidence of an association between HRT usage and the increased risk of IBC, particularly ILC, with combined oestrogen–progestogen HRT [40,42,43,94]. In the UK, it has been reported that up to 34 extra breast cancer diagnoses in every 1000 are due to HRT use, dependent on the duration and pattern of use [95]. There have been similar associations found between HRT use and in situ breast disease, with most studies showing a stronger association within LCIS than DCIS, and a stronger association with LCIS compared to ILC [42,43]. However, one US study found no association between the risk of DCIS and either ever-use or current-use of HRT in post-menopausal women [64].

## 3. Materials and Methods

### 3.1. Participants

The ICICLE and GLACIER studies were two UK-based case-control studies recruiting participants between 2007 and 2013, with the aim of investigating genetic predisposition to DCIS and lobular breast cancer, respectively (MREC references: 08/H0502/4 and 06/Q1702/64).

Cases were recruited to both studies at over 100 hospitals throughout the UK, and women aged 60 or younger at the time of diagnosis were eligible for the studies. For inclusion in the ICICLE study, cases had to be diagnosed with pure DCIS with no ipsilateral invasion >1 mm; for the GLACIER study, cases could be diagnosed with ILC, with LCIS alone, or LCIS concurrent with IBC. Controls were eligible for the study if they had no known family history of in situ breast cancer or IBC up to 2nd degree and also had no personal history of either in situ breast cancer or IBC. Cases identified control participants, as cases were asked to approach non-blood relatives and friends to act as controls and there were no age restrictions on controls.

Of the ~5900 cases, 293 from ICICLE and 180 from GLACIER were found to be ineligible after recruitment based on the histopathology of their first tumour and were, therefore, excluded from the case-control analysis. Additionally, 12 controls were excluded due to missing date of birth or not sending the blood samples required for genotyping. After exclusions, there were 2109 controls, and 5422 cases between the two studies, with the cases categorised according to their primary diagnosis into these subgroups; 3136 with DCIS, 1560 with ILC, and 726 with LCIS, of which 386 LCIS cases were alongside an ipsilateral invasive tumour of other morphology (mixed type or IDC).

For the purposes of this current study, only women aged ≥35 and ≤60 at diagnosis (for cases) or study entry (for controls) were included. The upper age limit was defined as 60 due to the ICICLE and GLACIER studies only recruiting cases aged 60 or under, although controls of any age were accepted. Additionally, the lower age limit of 35 was chosen as the controls had a different age distribution than the cases, and a small number of cases were younger than 35. Only the pure DCIS and LCIS cases were included, leaving 3075 DCIS and 338 LCIS cases, in addition to 1584 controls (Appendix A).

### 3.2. Data

As part of the ICICLE and GLACIER studies, all participants were asked to fill out a questionnaire at study entry, which included questions on past medical history, use of exogenous hormones, ethnicity, and reproductive history, which provided data on risk factors (Appendix A). For cases, histopathology reports were obtained from local hospitals in order to determine pathology data such as size, grade, and tumour receptors.

The number of births was calculated as the number of full-term births, including stillbirths, as reported by participants. Twins counted as one birth. Family history of cancer was considered as positive if 1st–3rd-degree relatives had reported a history of any cancer, and family history of breast cancer if 1st–3rd-degree relatives had a history of breast cancer. Where participants had not answered questions on childbearing/number of births or HRT use, these were considered not present (1.5% for number of births and 3% for HRT).

Women were asked questions on menstrual history as part of the questionnaire, and menopausal status was inferred from answers to these questions using the following rules: women who were still having periods were considered pre-menopausal, and women who reported that their periods stopped naturally or after removal of ovaries were post-menopausal. This was inferred based on methodology in the Million Women Study, which was a large prospective cohort study of nearly 2 million women in the UK, looking at breast cancer risk [96,97]. For those where periods had stopped after treatment related to their in situ breast cancer, after hysterectomy without ovary removal, or after use of hormonal contraceptives such as the Mirena coil or hormone injections, status was recorded as post-menopausal in women who were 53 or over at study entry, and pre-menopausal in women <53 at study entry. The age of 53 years was used as a cut-off in the Million Women Study, as 96% of participants were post-menopausal by that age.

Variables with missing data over 10% (age at menopause, length of HRT use, and length of oral contraceptive pill use) were imputed by multiple imputations using chained equations over 40 iterations. Variables used for the imputation included age at diagnosis, contraceptive use, HRT use, presence of rare genetic mutations, year of diagnosis, cancer morphology, parity, and age of menarche. Due to a lack of patterns in the data, missing data was imputed for most but not all missing data points. When attempting to impute the type of HRT taken, convergence was not achieved, and so this variable was not imputed. After multiple imputations, age at menopause had 1% missing, length of HRT use had 1% missing, and length of contraceptive use had 5% missing.

### 3.3. Analysis

Associations between in situ breast cancer groups and each of the risk factors were explored using individual logistic regression models compared to controls, adjusting by the age at study entry/diagnosis and then separated by inferred menopausal status. A case-only analysis was then undertaken to compare risk factors for DCIS and LCIS, using LCIS as the control group. A further multivariate logistic regression model was created, including the variables of interest that had displayed associations with either DCIS or LCIS. Logistic regression models were also run for the association between in situ breast cancer and oral contraceptive use, adjusted by year at diagnosis/study entry, and for the associations between in situ breast cancer and breastfeeding and HRT, stratified by age categories (Appendix A). In addition, as a secondary analysis, multinomial logistic regression models were run to investigate whether the association with contraceptives was related to histopathological subtypes of DCIS, and so DCIS was stratified by the oestrogen receptor status and nuclear grade (Appendix A). We also performed a sensitivity analysis for the associations with hormone replacement therapy and contraceptives, including only those diagnosed after 2005, due to changes in the use of both contraceptives [98] and HRT [99] in the UK (Appendix A). In addition, we matched cases and controls 1:1 on birth year (5-year intervals) and year of study entry (exact) to determine whether there were birth cohort effects present in this study population (Appendix A).

Age was considered as a confounder in all logistic regression models due to the likeliness of age being associated with multiple variables, such as contraceptive use, parity and number of births, menopausal status, and HRT use, and due to the difference in age distribution between the cases and controls. These logistic regression models were restricted to parous women only when looking at age at birth, breastfeeding, and the number of births, restricted to women who took HRT only when looking at the length of time taking HRT, and restricted to women who took the oral contraceptive (OC) pill only when looking at the length of time taking the OC pill. These logistic regression models were then presented as a forest plot. The “unknown” category of breastfeeding was removed from the forest plots due to wide confidence intervals and no significant association.

All analyses were performed using STATA/MP 17.0.

## 4. Results

The characteristics of 3075 DCIS, 338 LCIS, and 1584 controls are shown in Table 1. The median age of control participants was lower than in the DCIS or LCIS cases (*p* < 0.0001 on the *t*-test comparing all in situ cases to controls), and a higher proportion of DCIS cases were inferred as post-menopausal at study entry.

Table 2 shows the frequency of risk factors within the study participants before multiple imputations. A larger proportion of controls were nulliparous compared to the in situ breast cancer cases, although the median number of births was the same in the cases and controls. A smaller proportion of the controls took HRT (16.7%) compared to the DCIS or LCIS cases (~29%), while the frequency of oral contraceptive usage was similar across the cases and controls (~82%).

Logistic regression models were used to explore the associations between each of the risk factors and developing in situ breast cancer, initially considering all women and then in pre-menopausal and post-menopausal women separately. Each model for the different risk factors was adjusted by age at diagnosis for cases or age at study entry for controls. Logistic regression models were then run with cases only to assess differences in risk factor associations in DCIS and LCIS, with LCIS used as the control group. The results from the regression models for cases compared to controls are presented as forest plots in Figure 1, Figure 2 and Figure 3, Appendix A, and summarised below. The regression models stratified by menopausal status can be found in Appendix A.

There were statistically significant increases in the odds of both DCIS and LCIS with increasing age that the period stopped, with a 2% increase in risk per increase in year (DCIS OR: 1.02, 95% CI: 1.02–1.02, LCIS OR: 1.02, 95% CI: 1.02–1.03), but no association with age of menarche (Figure 1 and Figure 2). There was still no association between the age of menarche and in situ breast cancer after separating by menopausal status (Appendix A), and this was not associated more strongly with one form of in situ breast cancer over the other (Figure 3).

There was no evidence of an association between DCIS and parity, number of births, or age at first birth, either when looking at all cases or when separated by menopausal status. However, there was a borderline association between an increased risk of LCIS and an increasing number of births (*p* = 0.054), albeit no strong association with parity. In parous women, breastfeeding significantly decreased the risk of developing both DCIS (OR: 0.65, 95% CI: 0.54–0.77) and LCIS (OR: 0.63, 95% CI: 0.46–0.86) regardless of menopausal status. Breastfeeding after childbirth was not associated with one in situ breast cancer over the other, neither overall nor when separated by menopausal status. When looking only at the pre-menopausal DCIS cases, this association was of borderline statistical significance (OR: 0.77, 95% CI: 0.58–1.00, *p* = 0.053) but was significant in pre-menopausal LCIS, conferring 39% reduced odds of LCIS (95% CI: 0.38–0.97). Breastfeeding in parous women was significantly associated with reduced odds of both DCIS and LCIS in post-menopausal women.

HRT use increased the overall risk of DCIS by 24% (OR: 1.24, 95% CI: 1.05–1.46) and LCIS by 62% (OR: 1.62, 95% CI: 1.21–2.17), with even stronger associations in those who took long-term HRT (for 10+ years). HRT is sometimes offered to pre-menopausal women who are approaching menopause. Despite only a small proportion of pre-menopausal women taking HRT (134 participants, 7%), this was associated with an increased risk of both developing DCIS (OR: 1.82, 95% CI: 1.21–2.76) and LCIS (OR: 2.05, 95% CI: 1.09–3.86). In post-menopausal women, HRT use significantly increased the risk of LCIS (OR: 1.51, 95% CI: 1.08–2.11) but not DCIS (OR: 1.07, 95% CI: 0.89–1.28).

Long-term HRT use (10+ years) also significantly increased the risks of both DCIS (OR: 2.47, 95% CI: 1.51–4.04) and LCIS (OR: 4.75, 95% CI: 2.27–9.96) in post-menopausal women, albeit with a wide confidence interval in LCIS due to small numbers. Case-only analysis revealed that long-term HRT usage was more strongly associated with LCIS than DCIS (OR: 0.54, 95% CI: 0.31–0.94), and this association was further strengthened when restricted only to post-menopausal cases (OR: 0.49, 95% CI: 0.27–0.88). There were no associations between the type of HRT and developing either type of in situ breast cancer. A sensitivity analysis was conducted to address the potential differences in trends of exogenous hormonal use over time, excluding cases diagnosed before 2005. where, the association with long-term HRT use was still observed in both DCIS and LCIS (Appendix A).

There appeared to be a protective effect of using oral contraceptives in both DCIS (OR: 0.75, 95% CI: 0.62–0.90) and LCIS (OR: 0.72 95% CI: 0.51–0.99) compared to those not taking contraceptives. However, when the year of diagnosis was added to the logistic regression model (year of study entry with controls), this association was no longer significant in DCIS or LCIS (Appendix A). Additionally, in women taking oral contraceptives for >5 years, the direction of the estimates suggested an increased risk of DCIS in pre-menopausal women, which was not seen in LCIS, although the estimates were not statistically significant. The association with contraceptives was also explored, excluding women diagnosed before 2005, where the association with the use of the oral contraceptive pill was still associated with reduced odds of DCIS until adjusted by year of diagnosis (or year of study entry for controls) (Appendix A). To check that this association was not seen solely because of differences in age or because of a birth cohort effect, this analysis was repeated in cases, and controls matched 1:1 on the year of birth (5-year interval) and year of study entry (exact match). Here, the protective association was still observed until the year of diagnosis was included in the model (Appendix A).

A positive family history of breast cancer was also more strongly associated with LCIS over DCIS (OR: 0.77, 95% CI: 0.61–0.97), but this association was not significant when stratifying by menopausal status.

The associations were further explored in a multivariate logistic regression model, including breastfeeding, parity, length of time taking HRT, age at diagnosis/entry for controls, and age when periods stopped (Table 3). After adjustment, there was still evidence of a protective effect of breastfeeding on the risk of DCIS (OR: 0.67, 95% CI: 0.55–0.81) and LCIS (OR: 0.62, 95% CI: 0.45–0.86). Additionally, evidence of an association between long-term HRT and developing both DCIS (OR: 2.54, 95% CI: 1.54–4.20) and LCIS (OR: 4.29, 95% CI: 2.16–8.54) was also still present. However, in contrast to associations seen with IBC in the previous literature, parity seemed to increase the risk of both DCIS and LCIS.

When exploring associations with pathological subtypes of DCIS as a secondary analysis, the oral contraceptive pill did not show any evidence of an association with specific DCIS grades but did show an association with ER-DCIS, indicating that the contraceptive pill use was associated with reduced odds of ER- DCIS specifically (Appendix A). However, with only 50% of ER status recorded for DCIS cases, this analysis is likely of low power and prone to bias.

## 5. Discussion

This current study has shown that HRT use increases the risk of both types of in situ breast cancer, especially when taken for 10+ years and even when taken in pre-menopausal women. This supports previously found associations in IBC and in situ disease, with similar findings of stronger associations with the lobular subtype [94,100,101,102]. Comparing risk for DCIS and LCIS, we found that HRT had a much stronger association with LCIS with an OR of 4.62 (95% CI: 2.31–9.28) when taken for 10+ years, compared to an OR of 2.67 (95% CI: 1.66–4.30) for DCIS. Upon stratification of the association between HRT and in situ breast cancer by age, statistically significant associations were all still in the same direction, showing that this association is not due to age differences. However, with large amounts of missing data on the type of HRT taken, it could not be determined if there was a specific type of HRT that was more associated with increased risk. We showed for the first time that HRT use in pre-menopausal women is also associated with in situ disease, although these numbers were small. Furthermore, with statistically significant associations in post-menopausal women, women should be informed about the risks of taking long-term HRT and encouraged to reduce the duration of HRT use. Additionally, post-menopausal women who need to take long-term HRT should also be offered increased screening [95].

This is the first study to show a protective effect of breastfeeding against DCIS and LCIS, both when adjusted by age at diagnosis/study entry in parous women only, and on the multivariate analysis including HRT use, parity, and age period stopped. Additionally, when stratified by age, statistically significant associations were in the same direction, so this association is unlikely just due to the difference in age between cases and controls (Appendix A). Despite this association with breastfeeding, we found no evidence of an association between in situ disease and the number of births or age at first birth, in contrast to other studies of DCIS and IDC [66,103]. However, on the multivariate analysis, there were statistically significant increased risks of both DCIS and LCIS in parous women, which is in contrast to the protective effect seen with IBC. It is important to note that the study population was ~94% White European and that the associations found may not be extrapolated to other ethnic groups.

The lack of a protective effect with the number of births could be due to women in this study not having a high enough number of births to observe an association, as some studies have only seen associations of breast cancer in more than three children [36,40,63,69,70,71], and median number of births was two in this current study. However, the association between parity and in situ breast cancer risk in the multivariate model is in the opposite direction to what would be expected, which could possibly be due to artefact or chance.

For both DCIS and LCIS, there was no association with age at menarche and only a very small association with age at menopause for DCIS, supporting the findings of other studies of DCIS, but in contrast to IDC. DCIS and LCIS had very similar risk profiles, with only the length of HRT use and family history having a stronger association with LCIS than DCIS. The strong association of family history with LCIS was also observed by Claus et al. [104].

The finding of a protective effect of the oral contraceptive pill against in situ breast cancer was unexpected and is not in line with previous literature, which either shows a weak or no association. Interestingly, we did not see this association in pre-menopausal women, where we actually saw the opposite effect with pre-menopausal women taking the oral contraceptive pill for longer than 10 years having a borderline (*p* = 0.059) association with increased risk of DCIS, but not LCIS. The reason for the protective effect in post-menopausal women is not clear, but could be explained by potential confounding factors, bias, or artefact. As participants filled out questionnaires upon study entry and had to recall their past exposures, bias could affect this association, especially as cases may be more likely to remember medical history or risk factors related to developing breast cancer compared to controls. This is further supported by the slightly higher proportion of missing data on contraceptive use in controls. However, there was no other way to collect objective data on the risk factors included in this study. This association could also potentially be explained by differences in the oral contraceptive pill offered over time, which is supported by results in Appendix A showing a lack of significant association after adding the year of diagnosis into the logistic regression model. To explore this further, we performed a sensitivity analysis looking at the type of oral contraceptive, acknowledging the limitations due to the large amount of missing data on the type of contraceptive pill. This showed the protective association of contraception was only significant with the progestogen type contraceptives and DCIS, suggesting that the protective association seen could be due to trends in contraceptive prescription (Appendix A), as it has been shown that types of contraceptive use have changed over time [98], with progestogen only pills being used more frequently in recent years. Multinomial logistic regression models were added to determine whether this association was specific to histopathological subtypes of DCIS, where it was found that this protective effect was only present in ER-DCIS (Appendix A). However, due to the reported oestrogen receptor status in only 50% of DCIS cases, this analysis is likely to be low-powered and prone to bias. We explored this association with nuclear grade also, which only had 10% missing data but did not show any evidence of an association.

Despite adjusting for age in all of the models, due to the difference in age distribution between the cases and controls, it is possible that the study could be influenced by birth cohort effects. However, when investigating this by matching cases to controls 1:1 (Appendix A), the association with long-term HRT was still observed, showing that if there is a birth cohort effect present in this study, it is not large enough to solely account for the associations reported.

We chose to also analyse patients by inferred menopausal status, and this status was inferred based on the methodology used in the Million Women Study [96,97]. Some women may have been misclassified as post-menopausal, as it is hard to know the menopausal status for certain in the absence of testing for luteinising hormones and follicle-stimulating hormones. The decision to stratify the logistic regression models by menopausal status was made due to evidence of different associations between some risk factors, such as BMI and menopausal status [78,103,105,106,107]. As cases were recruited from more than 100 different hospitals throughout the UK, the study sample is likely representative of the in situ breast cancer cases in the population, although, on average, our cases were slightly younger than in other UK studies [87] due to the inclusion criteria of cases being aged 60 and under. Cases were both incident and prevalent cases, and controls were often friends/non-blood relatives of cases, which can increase response rates [108] but could suppress significant association through overmatching of some potential risk factors [109]. Controls were also only recruited if they had no family history of breast cancer, and having a family history may modify some lifestyle factors that are known to increase the risk of breast cancer. However, when we re-ran the analysis comparing controls to cases without a family of breast cancer, we found that the results did not differ significantly, with the majority of ORs in the same direction as presented here (results not included), including the protective effect of oral contraception.

From published literature, it would be expected that the risk factors predisposing to DCIS would be similar to those predisposing to IDC, but with less data on LCIS, there were not the same expectations of similarities with risk factors predisposing to ILC. We have shown that the risk factors associated with LCIS in our study were similar to those found previously for ILC, with a clear association with HRT and long-term use in post-menopausal women and breastfeeding in parous women. Unlike ILC, there was no association with age of menarche and no protective effect of increasing the number of births. However, with only 338 in the LCIS cases, it is possible that the study had low statistical power to detect some associations in the LCIS group, and important associations could have been missed. The smaller number of LCIS cases could be a consequence of LCIS being less frequent and possibly also due to LCIS diagnoses being mostly incidental findings, in contrast to DCIS.

## 6. Conclusions

This study has found some similarities in risk factors pre-disposing to both forms of in situ breast cancer and compared to established associations with IBC from previously published studies, e.g., with breastfeeding after childbirth and post-menopausal HRT use. It has shown that some risk factors are more strongly associated with LCIS over DCIS, such as length of HRT use and family history. It would be important to repeat these analyses in incident cases only and further examine the association between decreased risk of in situ disease and contraceptive use in post-menopausal women. However, with observed associations between increased risk of in situ breast cancer and HRT use, women should be informed of these risks with long-term HRT and encouraged to reduce the duration of HRT use. There is an argument for increased radiographic surveillance in women taking long-term HRT.

## Figures and Tables

**Figure 1 cancers-15-04397-f001:**
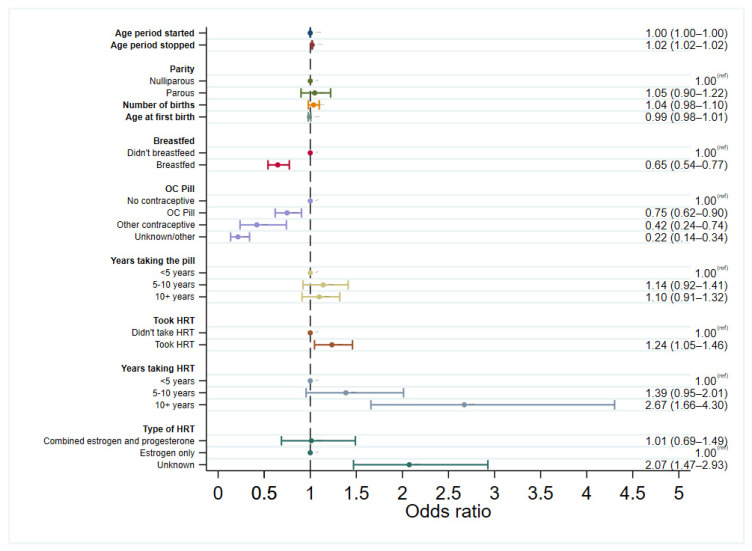
Logistic regression for developing DCIS vs. controls, adjusted by age (age at diagnosis for cases, age at study entry for controls).

**Figure 2 cancers-15-04397-f002:**
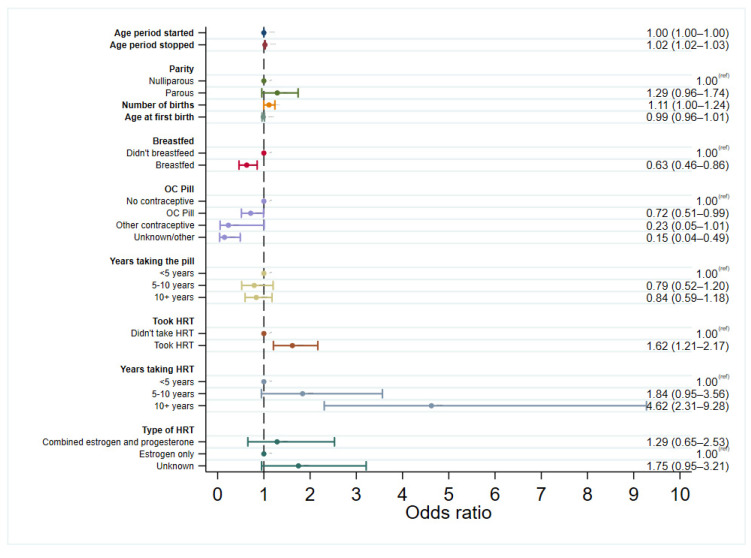
Logistic regression for developing LCIS vs. controls, adjusted by age (age at diagnosis for cases, age at study entry for controls).

**Figure 3 cancers-15-04397-f003:**
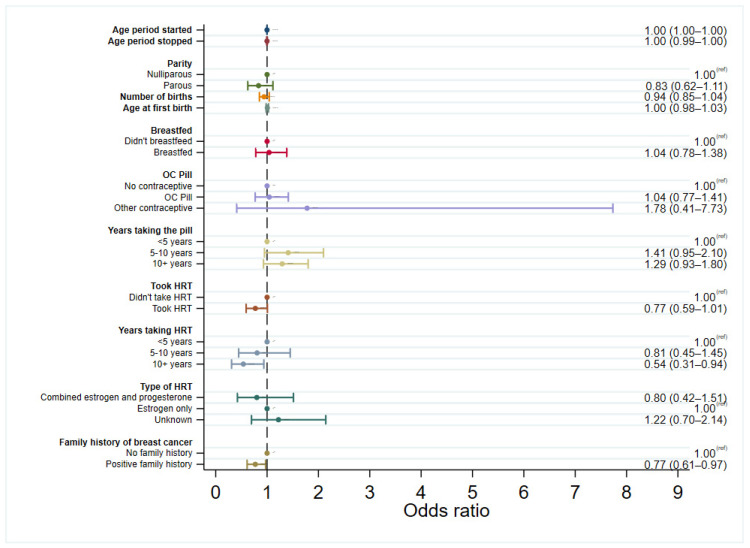
Logistic regression for odds of developing DCIS vs. LCIS, adjusted by age (age at diagnosis for cases, age at study entry for controls).

**Table 1 cancers-15-04397-t001:** Participant characteristics.

Characteristic	Controls	DCIS	LCIS
	*n* = 1584	*n* = 3075	*n* = 338
Age at first diagnosis (median, range) ^1^	48 (35–60)	52 (35–60)	51 (35–60)
Age at first diagnosis/study entry ^1^ (group)			
<50	888 (56.1%)	479 (15.6%)	71 (21%)
≥50	696 (43.9%)	2596 (84.4%)	267 (79%)
Inferred menopausal status			
Pre-menopausal Post-menopausal	894 (56.4%)	880 (28.6%)	141 (41.7%)
	690 (43.6%)	2195 (71.4%)	197 (58.3%)
Ethnicity			
Asian	22 (1.4%)	74 (2.4%)	3 (0.9%)
Black	8 (0.5%)	55 (1.8%)	1 (0.3%)
Mixed	9 (0.6%)	15 (0.5%)	5 (1.5%)
Other	5 (0.3%)	9 (0.3%)	0 (0%)
Unknown White European	60 (3.8%)	35 (1.1%)	7 (2.1%)
	1480 (93.4%)	2887 (93.9%)	322 (95.3%)
Time period of diagnosis ^1^			
<2000	0 (0%)	104 (3.4%)	35 (10.4%)
2000–2005	0 (0%)	385 (12.5%)	61 (18.1%)
2005–2010	794 (50.1%)	1581 (51.4%)	174 (51.5%)
2010–2015	790 (49.9%)	1005 (32.7%)	68 (20.1%)
Year of birth (median, range)	1962 (1947–1978)	1954 (1932–1977)	1955 (1929–1974)
Age at study entry/interview (median, range)	48 (35–60)	56 (35–77)	53 (37–79)
Year at study entry/interview (median, range)	2010 (2007–2013)	2011 (2007–2013)	2010 (2007–2012)

^1^ For control participants, this represents study entry, whereas in cases, this is the first diagnosis.

**Table 2 cancers-15-04397-t002:** Risk factors.

Characteristic	Controls	DCIS	LCIS
	*n* = 1584	*n* = 3075	*n* = 338
Parity			
Nulliparous	384 (24.2%)	642 (20.9%)	63 (18.6%)
Parous	1200 (75.8%)	2433 (79.1%)	275 (81.4%)
Number of births (median-range)	2 (0–4)	2 (0–4)	2 (0–4)
Age at first birth ^1^ (median-range)	26 (16–43)	26 (13–49)	25 (16–52)
Age at first birth ^1^			
≤25	512 (42.7%)	1154 (47.4%)	136 (49.5%)
>25	658 (54.8%)	1160 (47.7%)	131 (47.6%)
Unknown	30 (2.5%)	119 (4.9%)	8 (2.9%)
Breastfed ^1^			
No	222 (18.5%)	656 (27%)	72 (26.2%)
Yes	971 (80.9%)	1760 (72.3%)	202 (73.5%)
Unknown	7 (0.6%)	17 (0.7%)	1 (0.4%)
Age at menarche	13 (7–20)	13 (7–20)	13 (9–18)
Age period stopped ^2^	49 (24–58)	50 (17–64)	50 (26–59)
Contraceptive category			
No contraceptive	185 (11.7%)	519 (16.9%)	56 (16.6%)
Took OC pill	1302 (82.2%)	2493 (81.1%)	277 (82%)
Other contraceptive	31 (2%)	27 (0.9%)	2 (0.6%)
Unknown	66 (4.2%)	36 (1.2%)	3 (0.9%)
Years taking the pill ^3^			
<5 years	304 (23.4%)	593 (23.8%)	69 (24.9%)
5–10 years	255 (19.6%)	516 (20.7%)	44 (15.9%)
10+ years	492 (37.8%)	893 (35.8%)	91 (32.9%)
Unknown	251 (19.3%)	491 (19.7%)	73 (26.4%)
Took HRT			
Did not take HRT	1319 (83.3%)	2183 (71%)	241 (71.3%)
Took HRT	265 (16.7%)	892 (29%)	97 (28.7%)
Years taking HRT ^4^			
<5 years	123 (46.4%)	343 (38.5%)	35 (36.1%)
5–10 years	52 (19.6%)	204 (22.9%)	20 (20.6%)
10+ years	24 (9.1%)	179 (20.1%)	26 (26.8%)
Unknown	66 (24.9%)	166 (18.6%)	16 (16.5%)
Type of HRT ^4^			
Combined oestrogen and progestogen	78 (29.4%)	177 (19.8%)	26 (26.8%)
Oestrogen only	72 (27.2%)	161 (18.1%)	19 (19.6%)
Progestogen only	0 (0%)	16 (1.8%)	1 (1%)
Unknown	115 (43.4%)	538 (60.3%)	51 (52.6%)
Family history of breast cancer			
No	1584 (100%)	2038 (66.3%)	197 (58.3%)
Yes	0 (0%)	1037 (33.7%)	141 (41.7%)

^1^ Only in parous women; ^2^ only in post-menopausal women; ^3^ only in women taking the OC pill; ^4^ only in women taking HRT.

**Table 3 cancers-15-04397-t003:** Multivariate logistic regression model for in situ breast cancer.

Characteristic	DCIS vs. Controls	LCIS vs. Controls
Breastfed		
Did not breastfeed	1 ^ref^	1 ^ref^
Breastfed	**0.67 (0.55, 0.81)**	**0.62 (0.45, 0.86)**
Unknown	1.85 (0.74, 4.63)	0.62 (0.07, 5.26)
Years taking HRT		
Did not take HRT	0.81 (0.64, 1.02)	0.76 (0.50, 1.15)
<5 years	1 ^ref^	1 ^ref^
5–10 years	1.22 (0.82, 1.80)	1.40 (0.73, 2.68)
10+ years	**2.54 (1.54, 4.20)**	**4.29 (2.16, 8.54)**
Parity		
Nulliparous	1 ^ref^	1 ^ref^
Parous	**1.47 (1.18, 1.83)**	**1.94 (1.30, 2.87)**
Age period stopped	**1.02 (1.02, 1.02)**	**1.02 (1.02, 1.03)**
Age at first diagnosis	**1.06 (1.04, 1.07)**	0.98 (0.96, 1.01)

Results where *p*-value < 0.05 are highlighted in bold.

## Data Availability

Data supporting the results of this study is held by King’s College London.

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
