# Peer review of "DCIS and LCIS: Are the Risk Factors for Developing In Situ Breast Cancer Different?"

_cancers, 2023, doi:10.3390/cancers15174397_

Round 1

Reviewer 1 Report

Regarding 2.1.2  the triple negatives are associated with brca1 mutations but other genes have a wider spectrum.  the priority to test triple negative cancers only for mutations needs to be updated. CHEK2 are mostly ER-positive

does LCIS come in Er+ and ER- forms.  is this relevant?

line 178 typos

my main concern is regarding the matching of cases and controls.  this is not well detailed.  need to add to table year of birth, age at interview, year of interview. it appears that the oc effect goes away when year of birth is taken into account. Given the fluctuation in trends for HRT and OC use I think these need to be be better matched or show that the matching was sufficient to account for age and cohort effects.  Perhaps they could do a sensitivity analysis removing cases diagnosed before 2005.      

Reviewer 2 Report

Congratulations, his is a good work, with huge effort!

General comments

The article aims to summarize the risk factors for DCIS and LCIS and to analyse if they have similar associations to DCIS, LCIS and IBC. In general the paper is well written and the topic is of high interest for the medical practice and the women health. However, ORs are presented only for DCIS and LCIS compared to controls or for DCIS vs. LCIS, but not for IBC (invasive breast cancer, if my understanding is correct). An adjustment of the scope would be needed, for being in line with the results.

The methodology is accurately described. We speak about a case-control study, but performed on a large number of subjects (5900) and with the selection of controls some-how opportunistic. However, the limitations and potential biases are correctly described.

Some of the sections could be too detailed, which makes it too long and difficult to read (eg. Review of risk factors). A more synthetic style could be very useful for the readers.

Specific comments

Lines 276-277 – leaving the 3075 DCIS ... with controls – to rephraze for clarification

Section 3.1. Analysis – to include a short summary of all regressions that have been performed.

A figure with the study algorithm could be useful for the readers.

Section 4 (Results) – could be presented in a more synthetic way.

Conclusion – again, there is a reference to IBC (abbreviation for “invasive breast cancer” in introduction), which is not supported by the results.To review.

Reviewer 3 Report

The manuscript by Jasmine Timbres and co-workers analyzes similarities and differences among the  risk factors predisposing to DCIS and LCIS.

While the topic may be of scientific and clinical interest, the manuscript does not fully align with the goals of the special issue into which it was uploaded. There is in fact a discrepancy between the keywords of the special issue (solid tumors, targeted therapies, tumor microenvironment (TME), immune evasion, epigenetic changes, microRNA signature, metabolic features, precision therapy, drug resistance) and those of the manuscript (LCIS, DCIS, case-control, breast cancer, risk factor, logistic regression, breastfeeding, hormone replacement therapy, HRT).

In addition, the histopathological data regarding the DCIS nuclear grade and immunophenotype (expression of ER/PR, HER2, triple negative) as well as the subtype of ILCS (classical, florid and pleomorphic). deserve to be considered and investigated in order to identify more specific correlations. The multivariate analysis on these data could in fact explain in part what the authors have described regarding the use of oral contraceptives. In fact, some authors in the literature suggested that the use of the contraceptive pill could favor only high-grade forms of DCIS (Phillips LS, Millikan RC, Schroeder JC, Barnholtz-Sloan JS, Levine BJ. Reproductive and hormonal risk factors for ductal carcinoma in situ of the breast, Cancer Epidemiol Biomarkers Prev, 2009, vol. 18 (pg. 1507- t1514)10.1158/1055-9965.EPI-08-0967).

Table 1: Specify “White”.

Reviewer 4 Report

I was glad to review the work of the authors regarding this very interesting manuscript. The manuscript is well-written and the incorporated tables and figures make the study easy to follow.

Despite the major advances in breast cancer research, there are still numerous unanswered questions regarding the risk factors that have been previously shown to predispose to DCIS and LCIS, and  determine whether the risks associated with these risk factors are similar for DCIS and LCIS, and to IBC

1) A recently published article on the clinical significance of HER2 expression in DCIS is missing from your study

https://pubmed.ncbi.nlm.nih.gov/36352293/

I would suggest a brief discussion on this topic

Round 2

Reviewer 1 Report

I believe the cases and controls need to be matched for year of birth and year of interview.    they represent different cohorts  given the trends in parity, oral contraceptive use and HRT I would like to ensure that the cases and controls have the same year of birth and age at interview. the current version has too many times where the effect size was altered after consideration of age and year of birth. i would like to see a table where these variables are included and are shown to be similar across cases and controls   

Reviewer 3 Report

The authors took into account the previous comments and responded appropriately

Author Response

Dear reviewer,

Thank you for your response, we are grateful for your input.

Yours sincerely,

Ms. Jasmine Timbres and Prof. Elinor Sawyer (representing all authors; Kelly Kohut, Michele Caneppele, Maria Troy, Marjanka K. Schmidt, Rebecca Roylance).